# Directional Goldstone waves in polariton condensates close to equilibrium

Dario Ballarini [1]*, Davide Caputo[1,2], Galbadrakh Dagvadorj[3,4], Richard Juggins[3], Milena De Giorgi[1], Lorenzo Dominici[1], Kenneth West[5], Loren N. Pfeiffer[6], Giuseppe Gigli[1,2], Marzena H. Szymańska [3] & Daniele Sanvitto [1,7]

Quantum fluids of light are realized in semiconductor microcavities using exciton-polaritons, solid-state quasi-particles with a light mass and sizeable interactions. Here, we use the microscopic analogue of oceanographic techniques to measure the excitation spectrum of a thermalised polariton condensate. Increasing the fluid density, we demonstrate the transition from a free-particle parabolic dispersion to a linear, sound-like Goldstone mode characteristic of superfluids at equilibrium. Notably, we reveal the effect of an asymmetric pumping by showing that collective excitations are created with a definite direction with respect to the condensate. Furthermore, we measure the critical sound speed for polariton superfluids close to equilibrium.

[1] CNR NANOTEC—Institute of Nanotechnology, Via Monteroni, 73100 Lecce, Italy. [2] University of Salento, Via Arnesano, 73100 Lecce, Italy. [3] Department of Physics and Astronomy, University College London, Gower Street, London WC1E 6BT, UK. [4] Department of Physics, University of Warwick, Coventry CV4 7AL, UK. [5] PRISM, Princeton Institute for the Science and Technology of Materials, Princeton University, Princeton, NJ 08540, USA. [6] Electrical Engineering Department, Princeton University, Princeton, NJ 08540, USA. [7] INFN, Sez. Lecce, 73100 Lecce, Italy. *email: dario.ballarini@nanotec.cnr.it

Exciton-polaritons (hereafter polaritons) are bosons that condense in a driven-dissipative environment, where the steady state is achieved through a balance between gain and losses[1,2]. In this two-dimensional system, the pump is usually a non-resonant laser that creates a large population of excitons (in a reservoir), which quickly relax into lower-energy polariton states. Dissipation mainly occurs through leakage from the cavity mirrors, requiring constant feeding from the pump and allowing the optical detection of the polariton field by photoluminescence (PL) measurements. Above a density threshold, macroscopic quantum degeneracy of polaritons has been demonstrated in a variety of materials and structures up to room temperature[3–6]. However, the fundamental dynamics of the collective excitations of polariton quantum fluids are often hidden by the microscopic details of the disorder environment and by the effect of pumping and dissipation.

The excitation spectrum of polariton condensates is modified by drive and decay with respect to the equilibrium case of cold atoms, and is diffusive rather than linear at small momenta[7,8]. This is a general feature of driven-dissipative systems and prevents the Landau criterion from being fulfilled in non-resonantly pumped polariton condensates[9]. Recently, it has been suggested that the diffusive character of low-energy excitations can be strongly suppressed for long-lived polaritons (lifetimes longer than $100\,\text{ps}$) and further reduced if the polariton condensate is spatially separated from the exciton reservoir[10,11].

Although in principle the excitation spectrum of polariton condensates is accessible through PL experiments, in practice its detection is seriously hindered by the low signal-to-noise ratio, the strong emission intensity from the condensate itself and the relatively broad polariton resonances[12–15]. First indications of the existence of a soft Goldstone mode in time resolved experiments were obtained by the observation of a critical slowing down of the dynamics of an optical parametric amplifier[16,17]. More recently, four-wave mixing experiments have highlighted the presence of the ghost branch, which appears with negative energies with respect to the condensate[14,18–20]. However, so far any quantitative comparison has been challenging due to the short polariton lifetime and the small condensate size, limiting in energy and wavevector resolution a direct observation of the Goldstone spectrum.

In trapped atomic gases, Bragg scattering of two photons is used to detect collective excitations, allowing very accurate measurements of their energy and wavevector[21–23]. In the opposite limit of wave-particle duality, the Bogoliubov spectrum of photons in a hot atomic vapour has been recently measured by estimating the group velocity of transverse perturbations[24]. For quantum fluids of light, the excitation spectrum $S(\mathbf{k}, \omega)$ can be directly measured by performing the Fourier transform (FT) of the wavefunction in space and time, $S(\mathbf{k}, \omega) = |\text{FT}[\psi(\mathbf{r}, t)]|^2$[25]. A related technique is used to measure the dispersion of ocean waves by the FT of a time series of pictures taken from an aircraft: instead of measuring directly the water displacement, the light diffracted by the surface waves at different times is used to reconstruct the frequency-wavevector relation with high resolution[26].

In this article, we employ a high-quality semiconductor microcavity with a reduced density of defects and long polariton lifetime to form a condensate close to equilibrium and to measure the spectrum of its collective excitations by interferometric measurements of temporal and spatial coherence. This represents the optical analogue of the oceanographic technique used to obtain the dispersion of surface waves (Supplementary Note 1). In our case, the spatio-temporal oscillations of the polariton condensate are obtained through the fluctuations of the first-order correlation function. Using this technique we observe, for the first time, the transition to a linearised dispersion with increasing

particle density, showing the dominant phonon character of the Goldstone modes in a thermalised polariton condensate. Interestingly, collective excitations form with a preferential direction, much like surface waves in the ocean are oriented along the wind blowing in a specific direction. The origin of this peculiar effect in polariton condensates is due to the spatial displacement of the exciton reservoir from the polariton condensate, which results in an asymmetrically populated Goldstone dispersion.

## Results

**Measurements of the excitation spectrum.** The sample used in these experiments is a high quality factor ($Q > 10^5$) GaAs microcavity with a polariton lifetime of $\approx 100\,\text{ps}$ and twelve 7-nm quantum wells placed at the positions of maximum field enhancement within the cavity[27–29]. The pump laser (tuned to the first minimum of the stop band, at $E = 1690\,\text{meV}$, much higher than the polariton resonance) is focussed into a Gaussian spot with diameter $\approx 20\,\mu\text{m}$. After fast energy relaxation, a high density of excitons and polaritons is formed under the pump spot. As a consequence, an expanding cloud of polaritons ballistically propagate radially outwards from the injection point, while excitons, much heavier, are mainly localised within the pumping spot. In Fig. 1a, b, (showing momentum- and real-space cross sections, respectively) the $k$-delocalised emission at $E \approx 4\,\text{meV}$ comes from the high polariton density within the pump spot and the peaks at $|k| = 2\,\mu\text{m}^{-1}$ correspond to polaritons radially expanding from the pump spot outward (Supplementary Note 2). Above a critical density, energy relaxation occurs efficiently in the expanding polariton cloud through stimulated scattering into the bottom of the polariton dispersion, triggering the formation of a large polariton condensate with $k \approx 0$ all around the pumping spot[28,30,31]. The highest intensity peak (red rectangle) is measured at the bottom of the dispersion ($k = 0$) and is due to the emission coming from the extended condensate formed outside of the pump spot.

The signal coming from a region of the bottom condensate on the right side of the pump spot is sent to a Michelson interferometer with a corner reflector in one arm (R) and with a long delay line (M) in the other arm (Fig. 1c). An energy filter is applied to keep only the signal close (0.5 meV) to the bottom of the dispersion and avoid the contribution of higher-energy polaritons. The corner reflector is used to obtain the symmetric image around the central point (autocorrelation point), while the long delay line is used to measure the spatial correlations at different times. Selecting only one direction, the one dimensional interferogram is recorded for increasing time delays to obtain, from the fringe visibility, the two-dimensional spatio-temporal correlation map $g^{(1)}(\Delta x, \Delta t)$. The condensate density can be controlled by externally tuning the intensity of the exciting laser and, in the left column of Fig. 2, $g^{(1)}(\Delta x, \Delta t)$ is shown for increasing polariton density from top to bottom. In the right column of Fig. 2, the two-dimensional Fourier transforms of $g^{(1)}(\Delta x, \Delta t)$ directly show the momentum-energy relations with increasing densities from top to bottom.

For a weakly-interacting many-boson system without drive and dissipation, the role of interactions can be addressed in the Bogoliubov approximation by introducing new quasi-particles, defined as a superposition of condensed particles, corresponding to forward and backward propagating waves[32]. The associated Bogoliubov dispersion has a positive and a negative branch of the form:

$$\hbar\omega_{bog}(k) = \pm\sqrt{\frac{\hbar^2 k^2}{2m}\left(\frac{\hbar^2 k^2}{2m} + 2\mu\right)} \qquad (1)$$

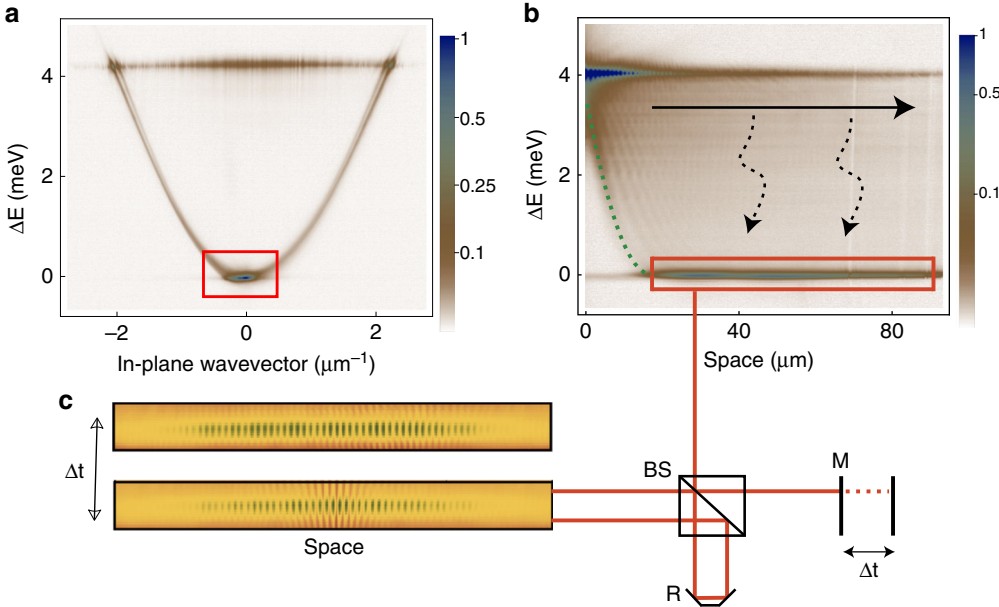

**Fig. 1 Measurements of spatio-temporal coherence. a** Momentum space PL showing the dispersion. The energy offset is $E = 0$ at the bottom of the polariton dispersion, corresponding to 1.6 eV. **b** Energy-resolved cross-section of real space PL. The pump is centred at $x = 0$ and the Gaussian profile of the energy blueshift ($\Delta E \approx 4$ meV), due to the high density of excitons under the pump spot, is indicated by the green, dashed line. The expanding polariton flow is indicated by the black arrow at $E \approx 4$ meV, corresponding to the emission at $|k| = 2$ μm$^{-1}$ in (**a**). The condensate is spatially separated from the pump and it is formed through energy relaxation (dashed arrows) of the polaritons ballistically ejected from the pump position. The region marked by the red rectangle at the lowest energy corresponds to the emission at $k = 0$ in (**a**. **c**). Interferogram for $\Delta t = 0$ and $\Delta t = 100$ ps obtained with a Michelson interferometer, where R is a corner mirror used to rotate the image around the autocorrelation point and a long delay line (M) has been included in the other arm.

where $m$ is the particle mass, $\mu$ the self-interaction energy ($\mu = g|\psi|^2$ with $g$ the particle-particle interaction strength and $\psi$ the condensate wavefunction) and $k = \frac{p}{\hbar}$ the wavevector of the matter wave. The effect of the out-of-equilibrium configuration, with pumping and decay, is instead reflected in the following spectrum of elementary excitations

$$\omega_{ee}(k) = -i\Gamma/2 \pm \sqrt{\omega_{bog}^2 - \frac{\Gamma^2}{4}} \qquad (2)$$

by the term $\Gamma$, which is a combination of pumping and dissipation rates[8,9].

Experimental data are compared in Fig. 3 with the dispersion relation of Eq. (2) and with the single-particle dispersion $\frac{\hbar^2 k^2}{2m}$. The single-particle parabolic dispersion is measured from the momentum space PL well below the threshold power for condensation (Fig. 3a). To take into account the small, but still measurable, velocity of the condensate that tilts the excitation spectrum with respect to the reference frame, the Doppler term $\omega_{dop} = (k - k_c)v_c$ is added to Eq. (2). Moreover, to center the dispersion at the energy and wavevector of the condensate, the angular frequency $\omega_{offset} = v_c k_c$ is used, giving a total frequency term of $\omega_{tot} = \omega_{dop} + \omega_{ee} + \omega_{offset}$. Approaching the threshold (Fig. 3b), the measured excitation spectrum (points are obtained from the positions of the maximum intensity in the data shown in the right column of Fig. 2) cannot be distinguished from the bare dispersion (dashed line). Linearisation can instead be seen in Fig. 3c and more clearly in Fig. 3d. Bogoliubov quasi-particles behave as phonons ($\omega = c_s k$, with $c_s = \sqrt{\mu/m}$ the speed of sound) at small momentum $p \ll mc$ and as $\frac{\hbar^2 k^2}{2m} + 2\mu$, i.e. the single-particle dispersion with an additional frequency shift due to interactions, in the opposite limit of large momentum. From Fig. 3c to d, the sound speed extracted from the fitting curves

increases from 0.35 μm/ps to 0.55 μm/ps and the healing length decreases from $\xi = 4$ μm to $\xi = 2.5$ μm, giving cutoff wavevectors of $\frac{1}{\xi} \approx 0.24$ μm$^{-1}$ and $\frac{1}{\xi} \approx 0.38$ μm$^{-1}$, respectively. The fluid velocity also increases, from $v_f = 0.04$ μm/ps in Fig. 3c to $v_f = 0.13$ μm/ps Fig. 3d.

In Fig. 4, the points of Fig. 3d close to the condensate energy (energy resolution is given by $\delta E = \hbar \frac{2\pi}{T} = 0.02$ meV, with $T \approx 200$ ps the total time delay in the interferometric measurements) and wavevector are shown zoomed in. The sound dispersion $\omega = ck$ follows the experimental data for $k < 0.3$ μm$^{-1}$, while for larger wavevectors the excitation spectrum tends to recover the parabolic dispersion (dashed-grey line). At all densities, $\Gamma < 0.005$ ps$^{-1}$, that is, the diffusive part of the excitation spectrum, where $\mathrm{Re}\,[\omega_{ee}(k)] = 0$ (flat dispersion), is limited to only very small wavevectors $k < 0.007$ μm$^{-1}$, i.e. much smaller than the momentum resolution of the measurements ($\delta k = \frac{2\pi}{L}$, with $L \approx 100$ μm the lateral size of the condensate). This means that the diffusive character of the excitation spectrum emerges only for distances larger than the condensate itself.

As can be seen in Figs. 2, 3, only one branch of the collective excitations, corresponding to forward propagating positive frequencies (and negative backward frequencies), is detected in the experiments. It is interesting to note that a finite condensate velocity only tilts the excitation spectrum in momentum space, and leaves the dispersion population symmetric. Our observations are instead very similar to measurements obtained for surface gravity waves in the presence of wind, for which only one branch can be observed (Supplementary Fig. 1). In our case, the directional driving force is ascribed to the localised pumping in an extended condensate, where we consider only a portion of the condensate placed on the right-hand side of the pumping laser. The quantum and thermal fluctuations which populate the collective excitations of the condensate develop within the pump

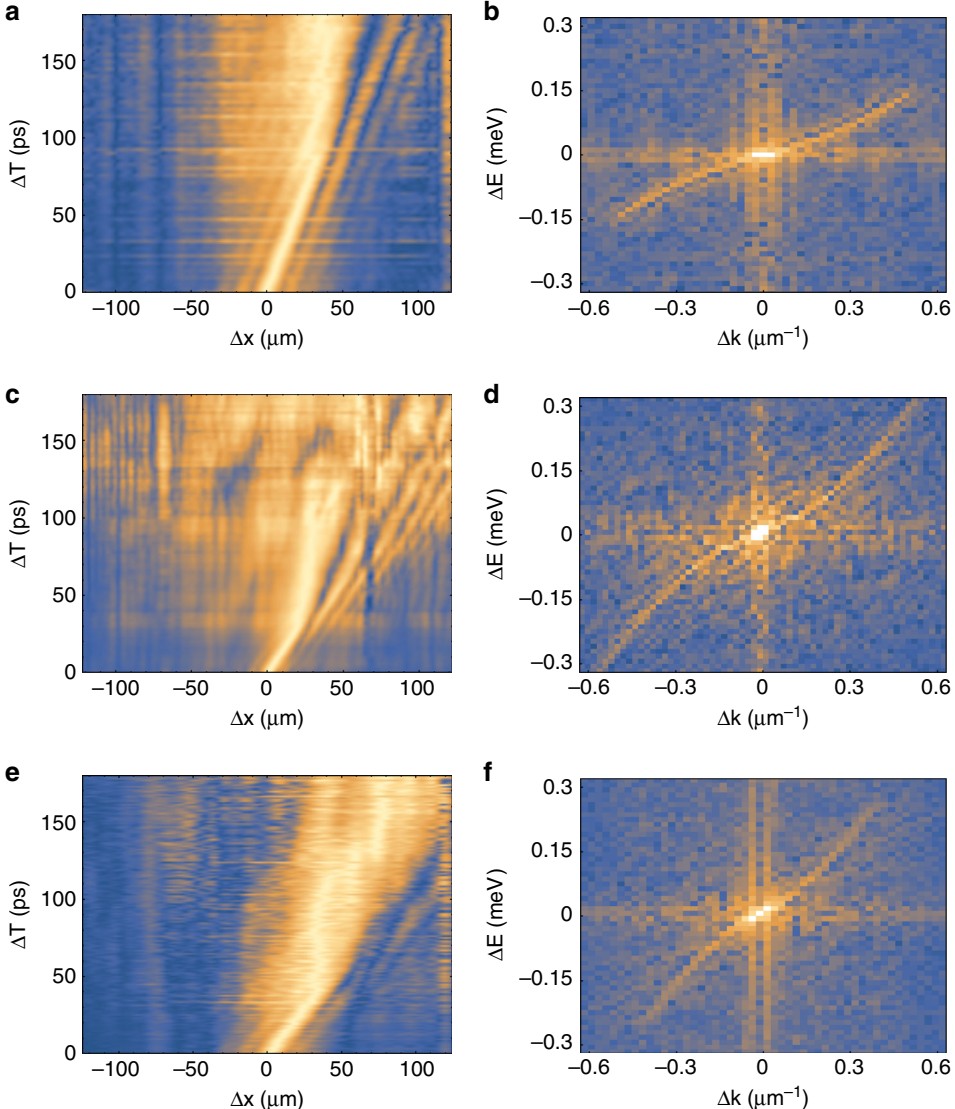

**Fig. 2 Excitation spectrum by Fourier Transform of correlation measurements.** First order correlation $g^{(1)}(\Delta x, \Delta t)$ in space and time for increasing pumping powers (from top to bottom). $P = 0.9P_{th}$, $P = P_{th}$ and $P = 1.5P_{th}$ in (**a**, **c**, **e**), respectively. **b**, **d**, **f** The Fourier transforms of (**a**, **c**, **e**), respectively.

spot region where the exciton reservoir is localised and are mostly directed radially outwards, resulting in a negligible "up-wind" contribution to the excitation spectrum in our measurements.

**Numerical simulations.** To fully understand why only one branch of excitations is populated, we investigate the system theoretically using multiple approaches (see Supplementary Note 3 for details). This is aimed at solving the main puzzle presented by the experimental observations: Is the asymmetry caused by the measurement technique, which uses $g^{(1)}(\Delta x, \Delta t)$ instead of $\psi(\mathbf{r}, t)$? Is it due to the non-zero wavevector of the condensate? Or is it the peculiar inhomogeneous driving set-up that gives the directionality to the excitations? We address the first two possibilities using both numerical and analytical approaches (see Supplementary Note 3), while the driven-dissipative inhomogeneous problem is suitable only to numerical analysis.

The numerical analysis utilises the truncated Wigner method[33] to simulate the polariton field $\psi$ with the same microscopic parameters as the experiments. Such an analysis is capable of recreating the experimental conditions and probing the effect of the inhomogeneous pumping. While the model used in this work does not take into account all the relaxation channels present in

the real systems, i.e. the contribution of phonons to the energy relaxation, this is not fundamental to the physics described as long as the observed population of the lower energy polariton states can be reproduced. Specifically, we calculate

$$i\hbar d\psi(\mathbf{r}, t) = dt \left[ -\frac{\hbar^2 \nabla^2}{2m} + i\frac{\hbar}{2} \left( \frac{\gamma(\mathbf{r})}{1 + \frac{|\psi(\mathbf{r},t)|^2_-}{n_s}} - \kappa \right) \right.$$
$$\left. + g|\psi(\mathbf{r}, t)|^2_- + V(\mathbf{r}) \right] \psi(\mathbf{r}, t) + \sqrt{\frac{\gamma(\mathbf{r}) + \kappa}{4}} dW \tag{3}$$

where $dW$ is the Wiener noise with zero mean satisfying $\langle dW^*(\mathbf{r}, t) dW(\mathbf{r}', t) \rangle = 2\delta_{\mathbf{r}, \mathbf{r}'} dt$, $m \approx 3.8 \cdot 10^{-5} m_e$ is the polariton effective mass, $\gamma(\mathbf{r})$ is the Gaussian pump, $\kappa \approx 1/200$ ps$^{-1}$ is the effective decay rate, and $g = 0.004$ meV $\mu$m$^2$ and $n_s = 1000$ $\mu$m$^{-2}$ are the effective polariton-polariton interaction strength and the saturation density, respectively. The Wigner commutator contribution has been subtracted from the polariton field density, $|\psi(\mathbf{r}, t)|^2_- \equiv |\psi(\mathbf{r}, t)|^2 - 1/2dV$, where dV is the size of the numerical grid. We have verified that for our parameters

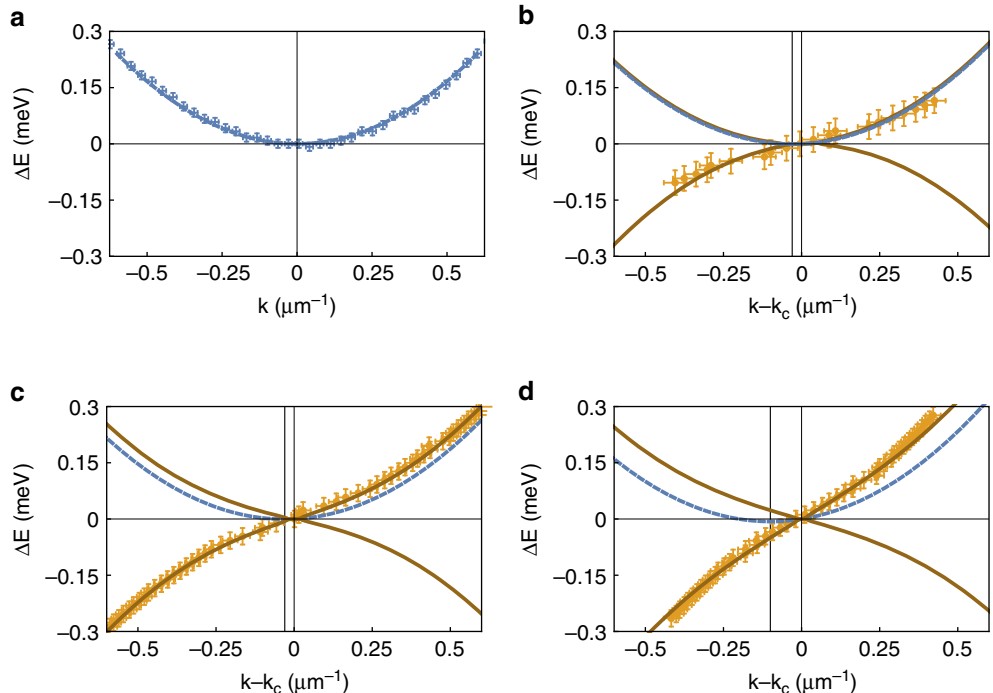

**Fig. 3 Dispersion relation for increasing densities. a** Parabolic dispersion extracted from PL at densities well below the condensation threshold $d_{th}$. **b-d** Excitation spectrum for increasing pumping powers corresponding to $P = 0.9P_{th}$, $P = P_{th}$ and $P = 1.5P_{th}$ as extracted from the positions of the maxima in Fig. 2b, d, f, respectively. Best-fit parameters to Eq. (2) are $\mu = 0.005$ meV, $\mu = 0.04$ meV and $\mu = 0.1$ meV in (**b-d**), respectively. The condensate wavevector is $k_c = 0.03$ μm$^{-1}$, $k_c = 0.03$ μm$^{-1}$ and $k_c = 0.1$ μm$^{-1}$ in **b-d**, respectively. The dashed-blue lines are the parabolic dispersion relations with the energy and momentum offsets given by $\mu$ and $k_c$ as obtained from the fit of Eq. (2) to the data in each panel. The mass is extracted from the parabolic fitting of the data in (**a**). $\Gamma = 0.005$ ps$^{-1}$ is used in (**b-d**).

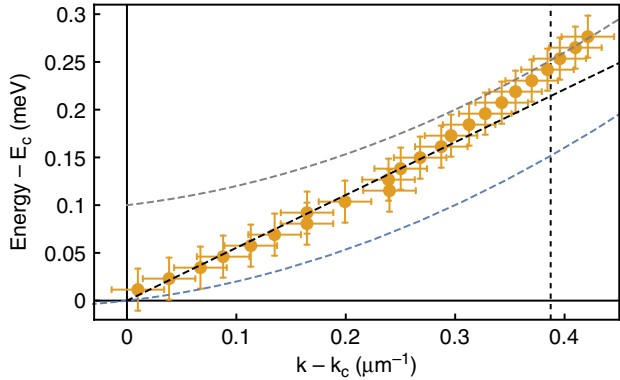

**Fig. 4 Linearised dispersion relation.** Experimental data of the excitation spectrum from Fig. 3d, zoomed in close to the condensate energy and momentum. The diagonal dashed-black line is the sound dispersion obtained from the same fitting parameters as used in Fig. 3d. Parabolic dispersion relations associated with the condensate energy and the high-momentum part of the spectrum are reported for clarity using dashed blue and grey lines, respectively. The vertical dashed line is the cutoff wavevector $\frac{1}{\xi}$ above which the dispersion is almost parabolic.

we are at low densities, $|\psi(\mathbf{r}, t)|^2_- \ll n_s$, and so have removed the associated correction from the noise term.

In addition to the $dW$ term, which gives the noise due to drive and decay, even in an equilibrium system thermal fluctuations can be included by adding white noise as an initial condition, and averaging over different realisations. We have performed numerics with both approaches, leading to qualitatively the same results. As the system is essentially thermal and, furthermore, we are only interested in the way the spectrum is populated rather

than the exact occupation of all the modes, the source of the noise is unimportant. For the results in Fig. 5, we used the initial condition approach to generate the noise.

Finally, in order to fully reproduce the experimental conditions, we add a wedge potential given by $V(\mathbf{r}) = \frac{V_0}{L_0}(L - x)$, where $V_0 = 0.1$ meV and $L_0 = 400$ μm, which is used in the simulations shown in Fig. 5. However, as demonstrated explicitly in the Supplementary Note 3, it is inessential to the physics observed here. The pump is located at the bright spot where the dashed lines cross in Fig. 5a, while we measure the condensate on the right-hand side of the pump (see also Fig. 5b, showing the phase of the condensate in 2D space with the region under consideration marked by a semitransparent rectangle). In the inset of Fig. 5a, we plot the polariton density along the horizontal and vertical cross sections passing through the pump, with a decreasing polariton density going from left to right in the region under consideration (indicated by the solid, red line). In Fig. 5c, we present the excitation spectrum obtained following the same procedure as for the experimental data, and showing that only one Bogoliubov branch is occupied, as in experiment. Note that the ghost branch appears populated in this case as a result of applying the Fourier transform to $g^{(1)}(\Delta \mathbf{r}, \Delta t)$, which is symmetric with respect to frequency. The true population of the ghost branch is shown in Fig. 5d, which displays the modulus squared of the FT applied to $\psi(\mathbf{r}, t)$ and the ghost branch appears here much less populated than in Fig. 5c (but still visible). Fig. 5d also shows a directionality of excitations, demonstrating that this effect is independent of the technique used in the experiments. In Fig. 5e, we show the PL obtained analytically in the Bogoliubov approximation for homogeneous (spatially uniform) pumping with finite condensate velocity (see Supplementary Note 3). The other parameters are the same as those in the inhomogeneously

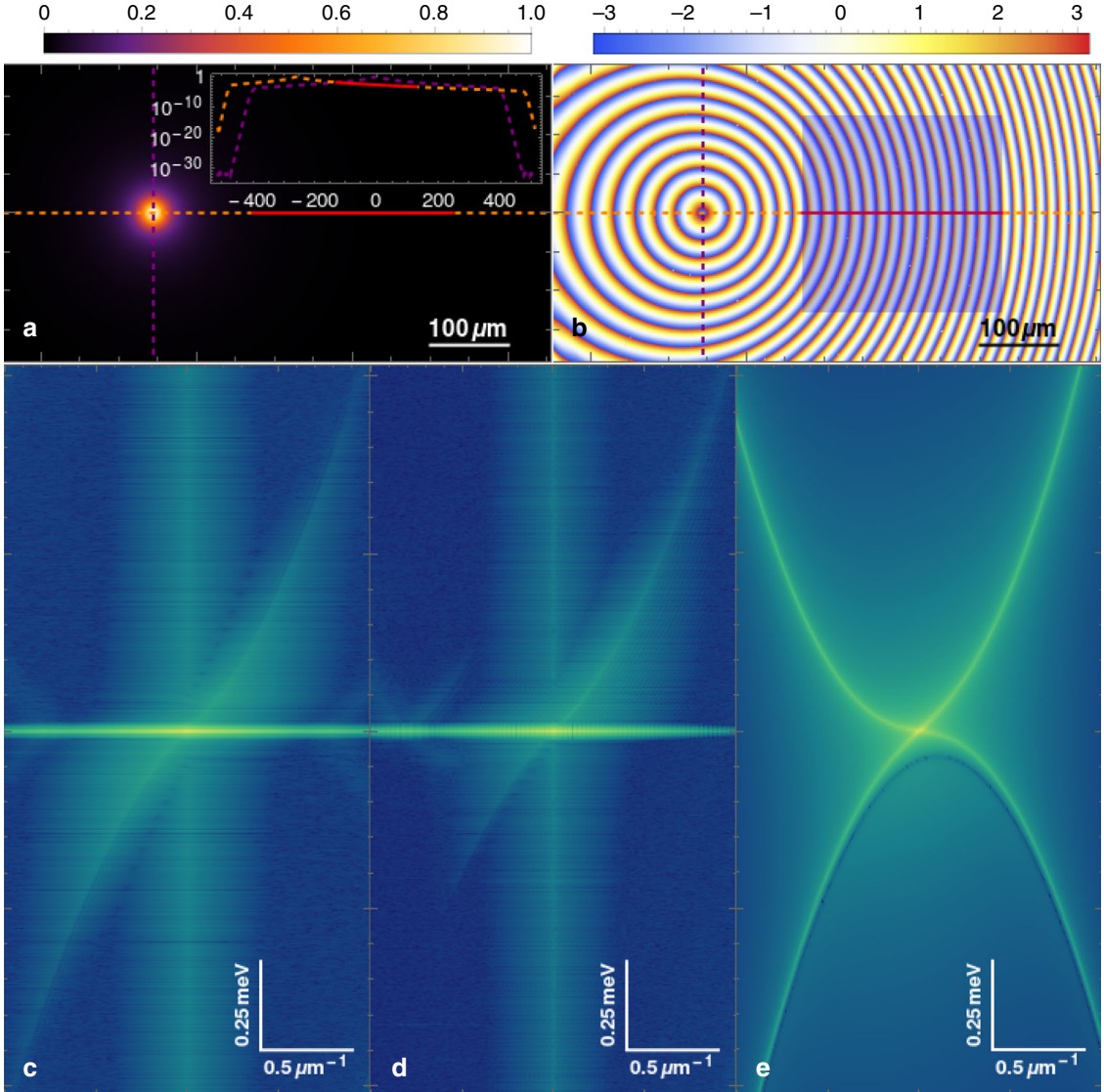

**Fig. 5 Numerical and analytical calculations of the spectrum and PL. a** Density profile of the condensate under the conditions of Fig. 1 with the pump placed on the left side of the measured region (solid red line in the horizontal profile). The main plot is on a linear scale and the inset a logarithmic one, showing the asymmetric polariton density of the condensate. **b** Phase (in radians) of the condensate in 2D space. The region under measurement is shown by the semitransparent rectangle. **c**, **d**, Excitation spectrum obtained numerically from the FT of $g^{(1)}(\Delta \mathbf{r}, \Delta t)$, i.e. following the same procedure as in the experiments, and of $|\psi(\mathbf{r}, t)|^2$, respectively. In the latter case the ghost branch is weakly populated, but still visible. **e**, PL from analytical calculations in the Bogoliubov approximation under uniform pumping with finite condensate velocity. Note that in this case both branches are populated.

pumped simulations, but nonetheless the directionality of the excitations is lost.

In conclusion, we have shown that the collective excitations typical of equilibrium BEC survive in a driven-dissipative environment, demonstrating the phonon-like character of a Goldstone mode. To obtain the frequency-wavevector relation of the collective excitations of the condensate, we have measured the fluctuations of the correlation function in space and time, in analogy to oceanographic techniques measuring the dispersion of surface waves. Moreover, we have investigated the origin of the asymmetry in the population of the excitation spectrum, finding that fluctuations originate in the region of the pumping spot and move away from the exciton reservoir, developing Goldstone waves with the same directionality. These results show that long-lived polaritons manifest universal features typical of interacting bosons at equilibrium and open the door to the optical investigation and control of the collective excitations in open-dissipative condensates by engineering the reservoir topology.

## Data availability

The data that support the findings of this study are available from the corresponding author upon reasonable request.

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

## Acknowledgements

The authors are grateful to I. Carusotto and I. Amelio for fruitful discussions and to P. Cazzato for technical support. This work has been funded by the POLAFLOW ERC Grant ID 308136 and the ERC project ElecOpteR grant number 780757. M.H.S. gratefully acknowledges financial support from EPSRC (Grants no. EP/R04399X/1 and no. EP/K003623/2). The work at Princeton University was funded by the Gordon and Betty Moore Foundation through the EPiQS initiative Grant GBMF4420, and by the National Science Foundation MRSEC Grant DMR 1420541.

## Author contributions

D.B., D.C., D.S., L.D., M.D.G. and G.G. performed the experimental measurements. G.D., R.J. and M.H.S. were responsible for the numerical and analytical results. K.W. and L.N.P. grew the sample.

## Competing interests

The authors declare no competing interests.
