## [Peer Review File · Nature Communications]

REVIEWERS' COMMENTS:

Reviewer #2 (Remarks to the Author):

The authors have addressed my concerns satisfactorily.

Reviewer #4 (Remarks to the Author):

The authors study both experimentally and theoretically the fluctuation spectrum of polariton condensates generated by a localized (Gaussian) non-resonant optical excitation. This localized excitation creates both a localized potential peak and a localized gain, which actually serves to create polaritons at two different energies. One distribution of polaritons is created with energy at the peak of the potential and is accelerated outward. A second distribution is created at low energy, just outside the potential peak region. The spatial and temporal coherence of this second distribution is detected and I assume that the higher energy polaritons are also coherent. Thus one can speak of two condensates, one at lower and one at higher energy.

The authors then consider the spectrum of fluctuations in the region to the right of the potential peak, finding that the spectrum (energy-wavevector plot) becomes tilted and that only the branch corresponding to fluctuations propagating to the right is populated. I believe this is fully reasonable. The upper energy condensate corresponds to polaritons propagating to the right and any fluctuations are affected by this condensate. Similar tilting was predicted in many theoretical works. As far as I know, the earliest was [I. Carusotto and C. Ciuti, Phys. Rev. Lett. 93, 166401 (2004)], where the fluctuations on top of a condensate created by a resonant pump with non-zero in-plane wavevector was considered. Furthermore, in the case considered by the authors, the fluctuations are most strongly created in the vicinity of the pump spot, which is consistent with the theoretical model used, where the Wigner noise is strongest in the pump region. According to the Bogoliubov theory of a spatially uniform condensate, these fluctuations would populate both branches and travel both to the left and to the right. However, the authors only observe the polaritons on the right-hand side, so most of the fluctuations observed should be those propagating to the right, corresponding to just the single branch (with right-ward group velocity).

I have read the reports of the three referees before me and also the responses of the authors to those reports. I agree with the authors that the ability to control the excitations on top of a condensate and modify their dispersion is interesting and an important step forward for the field. I do have a few comments of my own. However, as the review of this manuscript has already been going on for some time, I propose these as optional comments for the authors to consider. In any case, I agree with the positive comments of the other referees, and recommend publication in Nature Communications.

1) I find it surprising that the authors consider their system as close-to-equilibrium. Fig. 1 shows a very non-equilibrium distribution, with a significant number of polaritons appearing above the ground state energy. As I understood, the higher energy polaritons have an important influence on the lower energy ones and so should not be neglected? The first two paragraphs also highlight the relevance of the nonequilibrium nature of polaritons, so is this system not better seen as closer-to-non-equilibrium?

2) The authors mention obtaining the spectrum as the "Fourier transform (FT) of the intensity" and yet write $FT[\psi(r,t)]$. Is it that the Fourier transform of the amplitude is taken or do the authors mean $FT[|\psi(r,t)|^2]$?

3) The authors mention in the introduction that the Landau criterion can not be applied to driven-dissipative polariton condensates, yet later at the beginning of page 2 associate a linearized dispersion to a polariton superfluid. I have actually doubts that the present condensate should be called a superfluid. Possibly this is since the term superfluidity has been used interchangeably in previous works to describe the suppression of propagating polaritons with acoustic phonons or disorder. If we are referring to the lower energy condensate, then this is not propagating significantly as it corresponds to zero wavevector. If we are referring to the upper energy condensate, then I would've thought that scattering with acoustic phonons is still allowed and contributes to transfer of polaritons to the lower energy condensate. I don't see how scattering with disorder would be suppressed for either lower or upper condensate as the backward propagating states still exist.

4) The authors describe their pumping configuration as "asymmetric" and "off-axis", yet the pump is Gaussian (which is a symmetric function) and at normal incidence (according to the reply to Reviewer #2). In my opinion, their pumping configuration is actually symmetric, but it is rather the observation that is asymmetric: they look at the behaviour to the right of the pump spot. Probably looking at the left side will populate the opposite branch.

5) It is suggested that the higher energy condensate relaxes in energy as it is propagating (illustrated also by the arrows in Fig. 1b). However, the theory seems to represent a slightly different situation where polaritons are created from noise, most strongly in the vicinity of the pump spot. It might be worth commenting that the theory is not accounting for energy relaxation of polaritons that have already been generated, or moving the arrows in the figure closer to the pump spot, if it is believed that energy relaxation occurs strongest closest to the pump spot. I am not saying though that this would affect the conclusions. As far as I understand, it should not matter how the lower energy condensate is populated, so long as it is.

6) One sentence reads "However, Fig. 5d also shows the experimentally observed directionality of excitations, ...". Although I think I understood what the authors meant, this may cause confusion as Fig. 5d is a theoretical figure.

7) In the supplementary information, where the dispersion of water waves is quoted as depending on $\tanh(gD)$ I believe it should rather be $\tanh(kD)$, otherwise, the units of the quantities in the argument of the tanh function do not seem right.

Timothy C. H. Liew

Reviewer #4:

1) I find it surprising that the authors consider their system as close-to-equilibrium. Fig. 1 shows a very non-equilibrium distribution, with a significant number of polaritons appearing above the ground state energy. As I understood, the higher energy polaritons have an important influence on the lower energy ones and so should not be neglected? The first two paragraphs also highlight the relevance of the nonequilibrium nature of polaritons, so is this system not better seen as closer-to-non-equilibrium?

The referee is right in pointing out the out-of-equilibrium nature of the system. We highlight this aspect in the manuscript because the ability to tune the effect of pumping and dissipation is highly significant in open systems such as exciton-polaritons in microcavities. However, the peculiarity of this work is that the dispersion relation of the excitations of the lower energy condensate can be approximated, with a high level of accuracy, to that expected for a gas of bosons in thermal equilibrium. Our measurements are concerned only with the low energy modes of the system and the fluctuations directly above them [the high energy condensate is filtered out for all the measurements related to the $g^{(1)}$]. In this context, by close-to-equilibrium we mean that the low energy condensate exhibits a power-law decay of correlations with exponents smaller than 0.25 (the same for space and time) and a linearized excitation spectrum, which are well-known results valid for equilibrium condensates. Note that this is unusual, as ordinarily pumping and dissipation modify the correlation decay in space and time and the dispersion of excitations so as to be different to their equilibrium values. As noted by the referee, at high energies some non-zero non-equilibrium population exists. The system is not thermalised in the whole energy range shown in Figure 1, but it is thermalised (equilibrated) in the energy interval around the condensate in the ground state.

2) The authors mention obtaining the spectrum as the "Fourier transform (FT) of the intensity" and yet write $FT[\psi(r,t)]$. Is it that the Fourier transform of the amplitude is taken or do the authors mean $FT[|\psi(r,t)|^2]$?

We do indeed mean the Fourier transform of the amplitude and thank the referee for pointing this out. We have changed this to refer to the wavefunction and not the intensity.

3) The authors mention in the introduction that the Landau criterion can not be applied to driven-dissipative polariton condensates, yet later at the beginning of page 2 associate a linearized dispersion to a polariton superfluid. I have actually doubts that the present condensate should be called a superfluid. Possibly this is since the term superfluidity has been used interchangeably in previous works to describe the suppression of propagating polaritons with acoustic phonons or disorder. If we are referring to the lower energy condensate, then this is not propagating significantly as it corresponds to zero wavevector. If we are referring to the upper energy condensate, then I would've thought that scattering with acoustic phonons is still allowed and contributes to transfer of polaritons to the lower energy condensate. I don't see how scattering with disorder would be suppressed for either lower or upper condensate as the backward propagating states still exist.

As noted in the introduction, the Landau criterion cannot in general be applied to *driven-dissipative* condensates, which often have diffusive excitation spectra. Note that, in terms of the naïve Landau criterion, the low energy condensate in our work shows a linear excitation spectrum at low energies and therefore could be considered a superfluid, independently of whether it is propagating or not. In general terms, the fact that a condensate does not propagate does not mean it is not a superfluid.

One way to probe superfluidity is to make the condensate move past a defect, but another is to consider a moving object inside the condensate at rest ($k=0$). In the latter case even a condensate at $k=0$ could show no scattering from the moving object.

That being said, superfluidity is subtler than simply the phenomenon described by the Landau criterion, particularly in driven-dissipative systems such as polaritons (see for example the recent publications in Nature Communications, volume 10, Article number: 3869 (2019) and Nature Communications, volume 9, Article number: 4062 (2018)). To avoid any confusion, we have changed the word superfluid to condensate at the beginning of page 2.

4) The authors describe their pumping configuration as “asymmetric” and “off-axis”, yet the pump is Gaussian (which is a symmetric function) and at normal incidence (according to the reply to Reviewer #2). In my opinion, their pumping configuration is actually symmetric, but it is rather the observation that is asymmetric: they look at the behaviour to the right of the pump spot. Probably looking at the left side will populate the opposite branch.

We agree with the referee on this point – this is exactly what we meant in the work. Note that, as shown in the SI, the population of the excitation spectrum is indeed inverted at each side of the central spot. However, the important point here is that if the reservoir is displaced with respect to the condensate, the fluctuations possess a given directionality and move away from the reservoir. In the region under consideration, the pumping is therefore asymmetric, coming from one side of the condensate. Given the referee’s comment, we have rephrased in the text the sentences which can be misleading for the reader.

In the Introduction:

“The origin of this peculiar effect in polariton condensates is due to the asymmetric pumping configuration which results in an asymmetrically populated Goldstone dispersion.” -> “The origin of this peculiar effect in our case is due to the spatial displacement of the exciton reservoir from the polariton condensate, which results in an asymmetrically populated Goldstone dispersion.”

In the Results section:

“In our case, the directional driving force is ascribed to the asymmetric pumping configuration, where we consider only a portion of the condensate placed on the right-hand side of the pumping laser.” -> “In our case, the directional driving force is ascribed to the localised pumping in an extended condensate, where we consider only a portion of the condensate placed on the right-hand side of the pumping laser.”

“Moreover, we have investigated the origin of the asymmetry in the population of the excitation spectrum, finding that the off-axis pumping configuration is responsible for the directionality of the Goldstone waves as it leads to a breeze of quantum fluctuations blowing on the condensate.” -> “Moreover, we have investigated the origin of the asymmetry in the population of the excitation spectrum, finding that fluctuations originate in the region of the pumping spot and move away from the exciton reservoir, developing Goldstone waves with the same directionality.”

5) It is suggested that the higher energy condensate relaxes in energy as it is propagating (illustrated also by the arrows in Fig. 1b). However, the theory seems to represent a slightly different situation where polaritons are created from noise, most strongly in the vicinity of the pump spot. It might be worth commenting that the theory is not accounting for energy relaxation of polaritons that have

already been generated, or moving the arrows in the figure closer to the pump spot, if it is believed that energy relaxation occurs strongest closest to the pump spot. I am not saying though that this would affect the conclusions. As far as I understand, it should not matter how the lower energy condensate is populated, so long as it is.

In our theoretical analysis we do not describe the full microscopic details of the pumping process and how the polaritons are created. For example, we do not include phonon scattering channels from high energies down to lower states. In our case we simply create incoherent polaritons at all energies, which then can occupy the condensate and Goldstone branches. In our simulations, we include the inhomogeneity of the pump and this is enough to observe propagation in space of both the condensate and the noise. Due to the drive and decay parts, the energy is not conserved and can be lost when the noise and the condensate propagate into the lower energy part of the system, i.e. away from the pumping spot. We agree with the referee that the exact way that the low energy modes are populated does not matter (beyond requiring that the process is incoherent), and, following his/her comment, we have added at pg. 4 of the main text a sentence clarifying this point:

“While the model used in this work does not take into account all the relaxation channels present in the real systems, i.e. the contribution of phonons to the energy relaxation, this is not fundamental to the physics described as long as the observed population of the lower energy polariton states can be reproduced.”

6) One sentence reads “However, Fig. 5d also shows the experimentally observed directionality of excitations, ...”. Although I think I understood what the authors meant, this may cause confusion as Fig. 5d is a theoretical figure.

We thank the referee for pointing out that this may be confusing and have changed the sentence accordingly:

“Fig. 5d also shows a directionality of excitations, showing that this effect is independent of the technique used in the experiments.”